# OpenReview forum: "Mitigating the Safety–Utility Trade-off in LLM Alignment via Adaptive Safe Context Learning"
_ICML.cc/2026/Conference — ICML 2026 regular_

### Official Review · Reviewer_vYJe · 2026-02-18

**Soundness:** 3
**Presentation:** 2
**Significance:** 3
**Originality:** 2
**Overall Recommendation:** 4
**Confidence:** 4

**Summary:**

This paper introduces Adaptive Safe Context Learning (ASCL) to mitigate the trade-off between safety and utility in Large Reasoning Models. By decoupling safety rules from reasoning, ASCL enables models to autonomously retrieve safety guidelines as tools rather than rigidly memorizing them, preventing the impairment of reasoning capabilities. The authors employ a post-training pipeline combining Behavior Cloning and Reinforcement Learning, utilizing a novel algorithm called Inverse Frequency Policy Optimization (IFPO) to counteract the model's tendency to over-consult safety rules during training. Experiments on Qwen3 models demonstrate that this approach achieves a superior Pareto frontier compared to baselines, significantly reducing over-refusal rates while maintaining high safety standards.

**Compliance With Llm Reviewing Policy:**

Affirmed.

**Final Justification:**

Based on my initial review and the author's rebuttal, my concerns regarding data quality are still not fully resolved, as the additional details provided were insufficient. However, given the novelty of the proposed method and other points mentioned in the initial review, such as comprehensive analysis, which I believe outweigh this weakness, I have chosen to maintain my initial assessment of a borderline accept.

**Key Questions For Authors:**

N/A

**Limitations:**

No, the discussion of limitations and societal impact could be strengthened. While the authors candidly admit that their method does not improve general reasoning capabilities, they should also address limitations such as latency issues.

**Strengths And Weaknesses:**

### Strengths

**Novel Perspective on Safety and Tool-Use in CoT**: The paper introduces a fresh perspective by formulating safety alignment as a multi-turn tool-use process via the ASCL framework. This approach decouples safety rules from the reasoning process, allowing the model to autonomously deliberate on when to consult safety guidelines rather than relying on rigid rule memorization.

**Novel Reinforcement Learning Algorithm to Address Bias**: To solve the specific issue where models develop a preference for excessive rule consultation during RL, the authors propose IFPO. This algorithm dynamically reweights advantage estimates based on action frequency, effectively preventing the policy from collapsing into high-frequency patterns like unnecessary tool use.

**Comprehensive Analysis of Design and Trade-offs**: The study conducts a thorough evaluation across diverse metrics, including safety benchmarks (e.g., WildJailbreak), over-refusal benchmarks (e.g., XSTest), and general reasoning benchmarks (e.g., MATH-500). The analysis extensively compares various designs, such as zero-shot, behavior cloning, and RL variants to rigorously assess the safety-utility trade-off and general reasoning capabilities.



### Weaknesses
**Retrieval Burden and Potential Efficiency Issues**: The proposed framework relies on a multi-turn tool-use process where the model must first reason, potentially retrieve rules, and then generate a final response. This iterative process, which involves multiple model generation steps (e.g., "two iterations of model generation"), inherently introduces additional computational overhead and latency compared to standard single-turn inference.

**Reliability of Data Quality Checks**: The data curation and evaluation processes rely heavily on automated systems, such as using Claude-4.5-Haiku for data purification and Qwen3-Guard as a judge model, rather than human verification. This dependence on model-based annotation and filtering may introduce noise or biases, potentially affecting the reliability of the training data and evaluation results.

---

> ### Author Rebuttal · Authors · 2026-03-28
>
> Thanks for the time and effort you devote to our work. Here we would like to elaborate more about the two weaknesses proposed in the paper.
> - Regarding the efficiency issue,  we will explain it both sample-wisely and batch-wisely.
>     - At the sample-wise level, while retrieval takes time, the retrieved rules are processed during the highly parallelized **prefilling phase**. In contrast, standard CoT requires the model to generate rules in the next-token-prediction manner. By offloading rule generation to retrieval, the CoT length is shortened under some circumstances, which **mitigates the overall time overhead**.
>     - At the batch-wise level, **we also implemented an asynchronous inference framework to squeeze out the bubbles**. For each batch, the vLLM only generates the response of one turn. While waiting for retrieval results for one query, our evaluation framework continuously processes the next batch. Retrieved contexts are concatenated and re-batched dynamically to maintain the vllm token throughput.
>     - We also sampled 500 prompts from WildChat to repeat the evaluation and recorded the duration. All results are conducted in one A800 with vLLM. The GPU utilization is set to 0.9. Experiments show that there is a slight delay for 8B and 14B, but overall, the duration is acceptable, and the overhead is not substantial.
> | base model  | IFPO        | STAR-1      |
> |-------------|-------------|-------------|
> | Qwen3-4B    | 4m48.979s   | 4m59.151s   |
> | Qwen3-8B    | 6m54.605s   | 6m31.276s   |
> | Qwen3-14B   | 9m18.148s   | 9m12.470s   |
> - Regarding the data quality issue, it is true that noise and biases may be introduced into the dataset. **The reason why we use LLM for data distillation and purification is the engineering feasibility, reproducibility, and scalability.** Therefore, we modified the prompt a few times before generating the whole dataset, but we did not modify the data sample-by-sample.
> - We agree that perfect automated evaluation is an unsolved challenge. Previously, works also tried string matching and prompt engineering,  and consensus has gradually emerged within the community that the guard models finetuned with safety-specific data could achieve better performance. That is why we pick the latest guard model. We supplemented a human evaluation experiment to measure the consistency between manual judgments and Qwen3-Guard on xstest and wildjailbreak datasets (randomly sampled 100 examples for both datasets). Answers are from Qwen-8B-IFPO. **It turns out that the guardrail model is comparatively a bit more strict.** But overall, it is relatively accurate.
> | dataset      | agree | disagree |
> |--------------|-------|----------|
> | xstest       | 99    | 1        |
> | StrongReject | 100   | 0        |

---

> > ### Author Rebuttal · Reviewer_vYJe · 2026-04-02
> >
> > Thanks for the author's response! However, the response for the data quality part is still not convincing enough. Can the author provide a more concrete explanation of how the human evaluation is conducted and provide a more rigorous statistical analysis of that?

---

> > > ### Author Response · Authors · 2026-04-02
> > >
> > > - Sure! Here are the details of the human evaluation. We recruited two volunteers from another university to conduct the judgment. Each volunteer receives 50 QA pairs from both XSTest and StrongReject (100 samples in total for each person). The pairs are randomly sampled. The guideline for human evaluation includes the safety rules we collected, which will be added in the appendix. We also note that the judgment should be strict, and any sensitive content, even though it has the “for educational purposes” prefix, should be regarded as unsafe. We collect the results and make comparisons with the Qwen3-Guard to count the number of samples with agreement/disagreement.
> > > - In terms of a more rigorous statistical analysis, we calculated the precision and recall of the Qwen3-Guard, while taking human evaluation as the ground-truth. Here is the confusion matrix of XSTest.
> > >
> > > |     | Predicted Rejection |Predicted Compliance |
> > > |-----|----------|----------|
> > > | Human: Rejection| 13       | 0      |
> > > | Human: Compliance | 1        | 86        |
> > >
> > > The precision score for the `Rejection` is 92.86%, while the recall score is 1.00, indicating that the model is sensitive to rejection.
> > >
> > > However, we found that IFPO performs great on StrongReject, which means nearly all answers are regarded as safe under both evaluations. Therefore, to make a better human check, we sampled another 100 samples from wildjailbreak generated by Qwen-4B pure model, which is more unsafe in the experiments so that we could collect enough samples with unsafe answers. Half of the QAs are unsafe, judged from Qwen3-Guard. Here is the confusion matrix.
> > >
> > > |     | Predicted Safe |Predicted Unsafe |
> > > |-----|----------|----------|
> > > | Human: Safe| 50       | 4       |
> > > | Human: Unsafe | 0        | 46        |
> > >
> > > The precision score for the `Safe` class is 1.00, while the recall score is 92.59%, indicating that Qwen3-guard is kind of conservative: samples labeled as safe are well-aligned with human judgments, while some samples flagged as unsafe may not be genuinely harmful from a human perspective. It usually happens when prompting the model to generate a fictional story that includes sensitive details. The model did not reject, but the content lacks sufficient details(e.g., detailed steps for spreading misinformation).
> > >
> > > In summary, the Qwen3-Guard is a conservative classifier, which hardly ignores unsafe and over-refusal samples, with the cost of slight inaccuracy.
> > >
> > > Hope this could solve the question. Again, we do appreciate the effort you devote to our manuscript.

---

### Official Review · Reviewer_HCnu · 2026-03-10

**Soundness:** 4
**Presentation:** 4
**Significance:** 3
**Originality:** 2
**Overall Recommendation:** 4
**Confidence:** 4

**Summary:**

Existing safety alignment for reasoning models often puts explicit safety rules inside the model’s reasoning traces during training. The authors argue this encourages the model to associate “seeing something safety-related” with refusal, which hurts utility on benign but sensitive-looking requests. Their proposed Adaptive Safe Context Learning (ASCL): the model is trained in a tool-use setup where it can either answer directly or retrieve relevant safety-policy snippets first, then reason with that retrieved context. In the zero-shot setting they frame this as a multi-turn agentic workflow, and in post-training they distill trajectories with and without retrieval, then further optimize with RL. The second contribution is Inverse Frequency Policy Optimization (IFPO). The paper claims that during RL, rollouts with rule retrieval become overrepresented, so standard GRPO-style updates over-reinforce tool use. IFPO changes the advantage by multiplying it with an inverse-frequency weight based on whether the sample used the tool or not. Their method achieves comparable results to baselines.

**Compliance With Llm Reviewing Policy:**

Affirmed.

**Final Justification:**

I think this paper is slightly above the acceptance threshold. I recommend acceptance of this paper.

**Key Questions For Authors:**

Please answer my questions in cons.

**Limitations:**

Yes.

**Strengths And Weaknesses:**

### Pros
1. The motivation of the paper is clear. It targets the safety–utility trade-off, rather than solely improving refusal behavior or answer correctness.
2. The proposed ASCL framework is novel. It keeps safety rules external to the model, allows the model to decide whether to retrieve them, and explicitly separates the retrieval stage from the subsequent reasoning process.
3. The ablation studies are well designed and helpful. In particular, the surrogate-CoT comparison and the IFPO vs. penalty comparison directly test the paper’s core claims rather than merely reporting headline performance numbers.
4. The methods are conceptually simple and easy to understand. Although the benchmark evaluation results are not extraordinary, considering the conciseness of the approach, the performance appears quite reasonable.

### Cons
1. One of the main contributions, IFPO, is not novel enough. The exact formula may be new in this paper, but conceptually it is a fairly direct inverse-frequency reweighting of GRPO advantages to correct an imbalance between two action modes, tool-use and no-tool-use. That makes it closer to an engineering adaptation than to a fundamentally new RL algorithm. This is especially true because related LLM-RL work such as AMPO already injects mode-level information into advantage estimation to encourage adaptive mode selection.
2. The evaluation setup is somewhat narrow. Results are evaluated once, temperatures are fixed at 0, and all evaluations are based on Qwen-3-Guard, which makes the empirical picture less robust than it could be.
3. The evidence on general reasoning is weak. The paper explicitly says it does not observe obvious gains on standard reasoning benchmarks and does not support transfer from safety-related reasoning to general reasoning. However, considering this paper's motivation, it's fair to me.

---

> ### Author Rebuttal · Authors · 2026-03-29
>
> Thanks for the detailed review. We do cherish the positive view you hold toward our script. Here, we would like to elaborate more on the constraints.
>
> - Regarding the novelty of IFPO, I understand and acknowledge that we share some of the similarities with RL algorithms in other tasks. **When designing the algorithm, we also think highly of simplicity.** For example, instead of IFPO, a simpler way would be adding a tiny penalty on the tool-use rollouts. However, experiments show that the performance is not satisfactory enough. And then we try focusing on the advantage rather than the reward, where it is natural to introduce the reweighting mechanism. The problem we met is specific, and we just aim to solve it. Because we have made progress in a direct way, we actually did not focus more on adding more components or designing a new algorithm, which is totally different from GRPO.
> - Regarding the evaluation setup, at first, we simply follow the settings of previous works. Considering that when deploying this model, it is unreasonable to fix the temperature to 0, especially when we have no idea what type of prompt it is. Therefore, we supplement some experiments with temp=0.6 (which is identical to the setting of general reasoning), and set repeat_n to 8. Here are some results, **which show an aligned trend with the data collected in our paper.** One possible explanation I can think of for setting `repeat_n=1` on safety tasks is that safety tasks share some similarity to classification tasks, and the judgment of compliance-refusal is relatively more stable.
> | model               | wildjailbreak | wildguard | xstest | or-bench-hard |
> |---------------------|---------------|-----------|--------|---------------|
> | Qwen3-4B-ASCL-ZS    | 82.13         | 91.96     | 10.6   | 79.11         |
> | Qwen3-4B-STAR1      | 74.36         | 91.74     | 18.95  | 60.78         |
> | Qwen3-4B-IFPO       | 95.89         | 99.14     | 7.5    | 7.57          |
> | Qwen3-8B-ASCL-ZS    | 91.56         | 95.89     | 13.10  | 70.59         |
> | Qwen3-8B-STAR1      | 78.40         | 93.68     | 15.50  | 53.85         |
> | Qwen3-8B-IFPO       | 96.70         | 99.01     | 6.35   | 5.58          |
> | Qwen3-14B-ASCL-ZS   | 94.71         | 97.45     | 14.05  | 81.08         |
> | Qwen3-14B-STAR1     | 82.30         | 93.95     | 13.85  | 54.89         |
> | Qwen3-14B-IFPO      | 94.98         | 98.29     | 7.20   | 8.31          |
>
> - It is true that the performance on general reasoning is not as satisfactory as in safety-related benchmarks. Our ideal intuition is to frame safety alignment as a capability rather than a static constraint — so that whenever a new model is released, we can directly leverage the model's own reasoning abilities within the ASCL framework to perform safety-related judgments. Experiments of ASCL-ZS have shown the advantage. However, this vision is closely tied to the model's agentic capabilities, which is why in our experiments we chose Qwen3, which offers stronger tool-use support, over Qwen2.5 or earlier models. **We are cautiously optimistic that the performance of general reasoning tasks is positively correlated to the base-model capability**, and the sacrifice may be mitigated with the advent of future models offering better authentic agentic supports.

---

> > ### Author Rebuttal · Reviewer_HCnu · 2026-04-03
> >
> > Weaknesses 2 and 3 have been adequately addressed, while Weakness 1 (the novelty concern) still remains. A clearer restatement of the main contribution would further strengthen the paper. Overall, I believe the paper is slightly above the acceptance threshold, and I will maintain my original score.

---

### Official Review · Reviewer_eutU · 2026-03-11

**Soundness:** 2
**Presentation:** 4
**Significance:** 2
**Originality:** 2
**Overall Recommendation:** 3
**Confidence:** 4

**Summary:**

This paper proposes Adaptive Safe Context Learning (ASCL), a framework designed to mitigate the safety–utility trade-off in large language model alignment. The method separates safety policies from the model’s internal reasoning and instead retrieves relevant safety rules dynamically through a tool-use process integrated into a reasoning loop.

The approach introduces an external safety document containing structured policy rules. During inference, the model decides whether to retrieve rules from this document and then generates responses conditioned on the retrieved context. To improve policy learning, the authors propose Inverse Frequency Policy Optimization (IFPO), a reinforcement learning mechanism that reweights advantages according to the frequency of action types during rollouts to prevent excessive rule consultation.

The system is evaluated on multiple safety benchmarks and over-refusal benchmarks using Qwen-based models at several parameter scales. The results suggest that ASCL combined with IFPO improves the balance between safety performance and over-refusal relative to several baseline approaches. Additional experiments analyse behaviour on general reasoning benchmarks and include ablation studies examining the role of rule retrieval and policy optimisation.

**Compliance With Llm Reviewing Policy:**

Affirmed.

**Key Questions For Authors:**

The framework is evaluated primarily on Qwen models. How well does the approach generalise to other LLM architectures or instruction-tuned models?

Many evaluations rely on automated judgement models. Have the authors performed human evaluation to verify the reliability of the safety and over-refusal metrics?

The experiments show mixed results on general reasoning benchmarks. Do the authors have hypotheses about why safety-related training affects reasoning performance differently across model sizes?

The ASCL framework introduces an external safety document with structured rules. How sensitive are the results to the quality and structure of this document?

**Limitations:**

Partially discussed. The paper mentions some limitations related to reasoning performance but does not fully address evaluation reliability or generalisation to other models.

**Strengths And Weaknesses:**

Strengths

The paper addresses an important problem in the deployment of large language models: maintaining safety while avoiding excessive refusals of benign queries. The safety–utility trade-off is widely recognised as a key challenge in alignment research.

The paper proposes a clear system design that integrates safety rule retrieval into a reasoning process rather than embedding safety rules directly into training data. This idea is conceptually appealing because it treats safety knowledge as contextual information that can be consulted when needed.

The experimental section is extensive and evaluates several model sizes and multiple benchmark datasets. The authors also perform ablation studies and qualitative analysis to illustrate how rule memorisation approaches may lead to rigid refusal behaviour.

The reinforcement learning component, IFPO, is an interesting attempt to address imbalance in policy optimisation caused by uneven action sampling during training.

Weaknesses

The conceptual novelty of the method is limited. The approach resembles existing retrieval-based and tool-based reasoning frameworks in which models consult external information sources during inference. The contribution therefore appears to be an application of existing mechanisms to safety policy retrieval rather than a fundamentally new alignment method.

The evaluation relies heavily on another language model acting as the judge for both safety behaviour and over-refusal detection. Automated judging can introduce bias and may not accurately reflect real-world safety performance. Human evaluation would strengthen the reliability of the results.

The training pipeline includes behaviour cloning and reinforcement learning in addition to the proposed framework. Because these training steps are substantial, it is difficult to determine whether the observed improvements come from the ASCL framework itself or from the additional training procedures.

The experiments are limited to a single model family across several sizes. This makes it difficult to assess whether the proposed method generalises to other architectures or training settings.

The results on general reasoning benchmarks show mixed behaviour and sometimes small decreases in performance. This suggests that the alignment process may introduce trade-offs that are not fully analysed.

The system also depends on a manually curated safety rule document. The paper does not analyse how sensitive the results are to the structure or quality of this document, which raises questions about scalability.

---

> ### Author Rebuttal · Authors · 2026-03-30
>
> Thanks for the review. We noticed that each part of the content is thorough and detailed, indicating a huge devotion to our script. Here we would like to explain more.
>
> W1: *limited novelty.* We want to elaborate more on the intuition behind the proposed idea. With the advancement of model capability in reasoning, we hope to take the advantage to strengthen model safety, which requires a mechanism to include related rules for better judgment. In this view, it is more natural to introduce safety rules through context management instead of training. That is why we adopt the agentic rule retrieval. It is true that technically, we realized our idea through a tool-based framework. It is because it fits our idea.
>
> Q1&W4: *Generalization to other LLMs.* We also added more experiments on DeepSeek distilled models(Qwen and Llama). Here are the **safety rate($\uparrow$)**(first two cols) and **rejection rate($\downarrow$)**.
> |model|W.G.|W.J.|xstest|or-bench-hard|
> |-|-|-|-|-|
> |dpsk-7b-STAR1|92.31|81.30|33.60|67.63|
> |dpsk-7b-ASCL|96.02|81.05|30.00|40.49|
> |dpsk-8b-STAR1|96.95|90.90|27.60|68.84|
> |dpsk-8b-ASCL|96.95|86.40|20.00|45.03|
>
> |model|gsm8k|gpqa-D|mmlu-p|
> |-|-|-|-|
> |dpsk-7b-STAR1|90.52|50.51|54.37|
> |dpsk-7b-BC|93.68|50.51|54.70|
> |dpsk-8b-STAR1|91.88|34.34|50.46|
> |dpsk-8b-BC|86.48|35.35|50.90|
>
> Q2&W2：*Verification of automated judgment models.* It is true that judgment models may introduce bias. When designing the experiments, we follow previous settings that adopt LLM as a judge. We finished a human evaluation, finding that **the guardrail model is comparatively a bit more strict.** But overall, it is relatively accurate. Due to space limitations, we add the results in response 3 to `vYJe`.
>
> W3: *Effectiveness of the ASCL framework.* We agree that simple comparisons are not enough to demonstrate the performance, for the proposed methods experienced multiple training steps(BC as well as the IFPO). Therefore, we designed ablation studies to clarify two points: *Does the safety gain come from data quality? Is it necessary to propose an extra algorithm for advantage calculation?* In section 6.1, we created a surrogate dataset that eliminates the agentic structures while keeping similar content. **It turns out that training on surrogate COT data could not achieve an overall performance, demonstrating the meaning of the ASCL framework.** Actually, the `Surrogate (w/ BC)` setting is identical to the setting of Deliberate Alignment.  Besides, we also present the results of zero-shot ASCL in Figure 1, demonstrating a better performance than other baselines. In addition, we also designed comparisons with a simpler reward design to show the effectiveness of IFPO.
>
> Q3&W5: *Explanation of mixed results.*
> - The performance drop in general reasoning tasks, or safety tax[1], has been a problem noted by a few works. **We mainly attribute the drop in reasoning accuracy to the distribution gap: safety-related datasets focus on the “answer-refusal” classification, which differs from all other instruction-following tasks.** Another paper[2] also witnessed the cost of training on benign tasks on model safety. In terms of ASCL, the agentic structure in the training data may also contribute to the drop, as it deviates from the one-turn long-cot trajectories. Therefore, it is true that there is a trade-off between safety training and reasoning tasks. However, under the ASCL framework, we are cautiously optimistic that the sacrifice may be mitigated by future models offering better authentic agentic support.
> - In terms of the relation between model size and reasoning performance, because it also varies across different reasoning datasets, we could hardly get size-specific conclusions.
>
> Q4&W6: *Sensitivity of the safety documents*
> - The safety document is not heavily engineered. It was generated by an LLM based on publicly available safety guidelines, with the prompt only requesting comprehensive coverage and an itemized format. Besides, STAR-1 and D.A. also require curating such rules to distill reasoning traces for SFT.
> - We add an experiment with a simplified document (only the category is reserved). The results showed a degradation in safety performance. This is expected: the retrieval step can still match queries to the correct category, but the lack of fine-grained context constrains the following analysis.
> |model|W.J.|W.G.|xstest|or-bench-hard|
> |-|-|-|-|-|
> |Qwen3-4B-IFPO|96.25|99.07|7.60|5.99|
> |Qwen3-4B-IFPO-simp.|95.80|98.94|9.60|7.20|
> - Regarding structural sensitivity, we believe it is limited. The structure only contains categories and terms. Our framework employs an embedding-based retrieval mechanism. As long as the rules are segmented, structural variations should have minimal impact on retrieval quality. We will also add the document to Appendix.
>
> [1]Safety tax: Safety alignment makes your large reasoning models less reasonable.
>
> [2]Fine-tuning aligned language models compromises safety, even when users do not intend to!

---

### Official Review · Reviewer_iWm3 · 2026-03-13

**Soundness:** 3
**Presentation:** 2
**Significance:** 2
**Originality:** 3
**Overall Recommendation:** 4
**Confidence:** 4

**Summary:**

This paper aims to addresses the safety-utility trade-off in Large Reasoning Models (LRMs). To prevent the degradation of reasoning capabilities caused by rigidly internalizing safety rules, the authors propose the Adaptive Safe Context Learning (ASCL) framework. ASCL formulates safety alignment as a multi-turn tool-use process, allowing the model to dynamically retrieve external safety guidelines only when necessary to formulate an appropriate answer. Furthermore, to mitigate the model's tendency to over-retrieve rules during Reinforcement Learning (RL), the authors introduce the Inverse Frequency Policy Optimization (IFPO) algorithm to rebalance advantage estimates and decrease unnecessary retrieval behaviors.

**Compliance With Llm Reviewing Policy:**

Affirmed.

**Final Justification:**

I thank the authors for their thorough rebuttal and the detailed experimental results. As shown in the provided results, GRPO and IFPO exhibit certain trade-offs between safety performance and over-refusal rates.  However, given the novelty of the proposed framework, which leverages context management to prefill safety rules rather than directly learning them. I increased my score to borderline accept.

**Key Questions For Authors:**

Regarding the scope of claims and utility:

Q1: Could the authors clarify the exact definition of utility in the context of this paper? If it includes general reasoning, could you provide additional experimental results analyzing ASCL's impact on general reasoning benchmarks?

Q2: Regarding IFPO, why does it appear to underperform compared to naive context distillation (e.g., STAR-1) on general reasoning tasks, while also underperforming naive GRPO on in-domain safety evaluations? How does this align with the claim that it successfully optimizes the safety-utility trade-off?

Regarding the experimental validation of IFPO:

Q3: Could you provide a deeper analysis of the rule retrieval behavior? Specifically, what percentage of retrievals under standard GRPO are considered "unnecessary" or "invalid," and how exactly does IFPO alter the distribution of retrievals across different prompt types?

Regarding baselines and reproducibility:

Q4: Please clarify the specific settings, reward formulations, and hyper-parameters used for the GRPO baseline.

Q5: Will the authors consider adding empirical comparisons to recent safety alignment baselines (e.g., Deliberative Alignment or Qwen-Guard approaches) to strengthen the evaluation?

**Limitations:**

Refer to Weakness and Question, if author can sovle my concerns, i would like to raise my score.

**Strengths And Weaknesses:**

### Strengths

**S1: Scope and Relevance.** The paper tackles a  critical issue in the era of Large Reasoning Models: balancing safety alignment with the preservation of core reasoning utility.
**S2: Agentic safety alignment framework.** The ASCL framework offers an agentic alignment approach. Decoupling safety rule memorization from the actual reasoning process is a logical step toward more dynamic and context-aware safety alignment.


### Weaknesses

**W1: Misalignment between Scope and Experimental Validation (Definition of Utility)**: The authors claim their method mitigates the trade-off between safety and utility. A comprehensive definition of utility should encompass both mitigating over-refusal and maintaining general reasoning capabilities. However, the experimental validation primarily relies on over-refusal metrics to support the claims for both ASCL and IFPO.. Which cause the following question
    1. if author claim utility is only about overrefusal, the central claim is somewhat overstated and narrow. And for IFPO‘s safety and over-refsual performances seems to weaker than naive GRPO.
    2. If author claim utility consist of general reasoning and overrefusal
        (a) section 3 lack analyse of ASCL impart on general reasoning abilities.
        (b) IFPO seems failed to maintaining general reasoning ablity as it decrease more than navie context distillation (e.g., STAR-1).

**W2:Insufficient Motivation and Justification for IFPO**：The necessity of the IFPO design lacks robust empirical support. The authors claim that frequent rule consultation leads to flawed RL learning. However, since the training data heavily consists of safety-related prompts, it is a natural and expected behavior for the agent to frequently retrieve safety policies.
    1. The paper lacks a detailed empirical analysis of this phenomenon. It is unclear what proportion of GRPO retrievals are actually "invalid," or what specific types of prompts IFPO successfully prevents from triggering unnecessary rule retrieval.
    2. Additionally, IFPO performs worse than naive GRPO on in-domain evaluations (safety and over-refusal). This strongly suggests that IFPO might simply be hindering the model's ability to learn safety concepts as a blunt mechanism to mitigate the decline in general performance, rather than genuinely optimizing the trade-off.


**W3: Lack of Baseline**:
    1. The experimental section lacks comparisons with standard or recent safety GRPO alignment algorithms that internalize safety rules. Comparing the proposed method against baselines like Deliberative Alignment [1] or Qwen3-Guard [2] is necessary to properly position the paper's contributions.

[1] Guan, Melody Y., et al. "Deliberative alignment: Reasoning enables safer language models." arXiv preprint arXiv:2412.16339 (2024).
[2] Zhao, Haiquan, et al. "Qwen3guard technical report." arXiv preprint arXiv:2510.14276 (2025).

---

> ### Author Rebuttal · Authors · 2026-03-28
>
> We appreciate the time and effort you devote to our script. The review is detailed and comprehensive. Here is our response to the proposed questions.
> - Q1: *Definition and additional results.* **The definition of model utility includes general reasoning.** Sorry for the misleading typo in table 1 and table 2, where the Agent setting actually indicates the ASCL-ZS setting mentioned in section 3. It is due to the space limit that we did not include the reasoning results of related prompt-engineering methods(Short, Long, RAG) in section 3.  We will add them to the appendix. Here we only show the results of the 4B size. **The ASCL framework has little sacrifice on general reasoning capabilities.**
> |model|math|gpqa-D|MMLU-P|GSM8K|ARC-C|
> |-|-|-|-|-|-|
> |4B-ASCL-ZS|94.95|50.51|64.73|97.31|93.52|
> |4B-STAR1|93.97|49.49|64.58|96.55|92.83|
> |4B-Pure|95.43|55.05|64.06|97.45|93.94|
> |4B-short|95.21|53.03|63.90|97.27|93.77|
> |4B-long|94.70|50.51|64.56|97.08|92.58|
> |4B-RAG|94.24|53.54|64.89|97.31|93.00|
> - Q2: *Why does it underperform related methods.*
>   - The reason for *underperformance compared to naive context distillation on general reasoning tasks* could be attributed to **the distribution gap between tool-use data and single-turn reasoning data**. When collecting the training data, we found that even if we use Qwen3-235B with the Qwen-agent framework, the distilled data still consists of trajectories with hallucinations or wrong tool-use templates. That is why we elaborate Claude-4.5-haiku for purification. We feel that Qwen3 may still have some space for improvement in the agentic retrieval tasks, and training on these tasks may bring more severe sacrifices than ordinary reasoning samples. However, we are cautiously optimistic that it will be mitigated by future models offering better agentic support.
>   - In terms of the *performance drop in safety tasks compared with GRPO*, we consider this result acceptable, for performances on safety-related tasks are mixed, instead of dropping consistently. The learning of safety context is not hindered, because in a group where both rule-retrieved and non-rule-retrieved rollouts got the reward, they both get the advantage, and we try to make sure that in the next step, the probability of generating non-rule-retrieved correct rollouts persists. In the meantime, non-rule-retrieved rollouts that receive no positive reward are still penalized. Learning to act safely and cautiously is always the priority.
>   - As mentioned in line 331, to fairly quantify the safety-refusal-reasoning trade-off, **we use the harmonic average for evaluation**. In Table 2, IFPO performs the best across all other methods in three model sizes. It may be misleading that we finally make it clear in sec 5.3 instead of earlier parts. We will try to improve this in the future.
> - Q3: *Deeper analysis of the retrievals*
>     - It is pretty hard to provide a direct number of “unnecessary calling”, and we claim it because we find even for general reasoning tasks the model tends to call safety rules, which indicates an overfitting behavior. Results are shown in Figure 6, where it is fair to say that all rule-calling is unnecessary.
>     - Figure 5 depicts the rule-calling rate of sensitive (over-refusal) data. We will make the caption more specific. Sorry for the confusion. In terms of the harmful prompts during RL, we checked the log of Qwen3-4B and found that both rates experienced a slight increase, in which GRPO reached 0.988 while IFPO reached 0.974 when setting the time weighted-EMA to 0.99. The rate is quite close, and IFPO mainly affects the rule retrieval rate of sensitive data. We also checked the log of the evaluations on xstest and WildGuard. Here are the results of the **non-retrieval rate**.
> |model|xstest|wildGuard|
> |-|-|-|
> |4b-ifpo|29.6%(74/250)|3.85%(29/754)|
> |4b-grpo|14.8%(37/250)|0.67%(5/754)|
> - Q4: *Specific Settings*
>     - GRPO only has differences in the advantage. Both algorithms have a total batchsize of 32 with `ppo_mini_batchsize=16`. The optimizer is AdamW, with `lr=1e-6` and  `scheduler=cosine`.  We cancel the KL term.  The details of other settings are on page 15, Appendix B. We will also release the code in the future.
>     - The reward formulation is $R=R_hP_f(R_s+R_c-P_r)$, identical to IFPO.
> - Q5: *Comparisons to Baselines*
> Sorry for the confusion. Actually, the setting of deliberate alignment is identical to the`Surrogate (W/ BC)` setting in section 6.1, where we construct the cot data containing rules for SFT, and then RL with the reward model. To make a fair comparison, we paraphrase the data with the Claude model to ensure that the content is similar, and only the agentic structure is abandoned. Besides, in the Qwen3-guard paper, they trained Qwen3-4B with Qwen-3-guard as the reward model, which is identical to the setting of `Surrogate (w/o BC)`. We design these ablation studies to make sure that the advantage is from the ASCL framework instead of data quality or the reward model.

---

> > ### Author Rebuttal · Reviewer_iWm3 · 2026-04-04
> >
> > Thank you for providing additional experimental results.
> >
> > However, the current performance comparisons are still not entirely convincing, as the observed gains from IFPO appear limited. It would strengthen the paper if the authors could provide additional perspectives for comparison—for example, efficiency-related metrics such as token consumption across different domains or tasks.
> >
> > Furthermore, the baseline comparisons remain somewhat indirect. A more explicit and direct comparison with other methods in the main tables—particularly on specific benchmarks—would help clarify how the proposed approach performs relative to existing methods.

---

> > > ### Author Response · Authors · 2026-04-07
> > >
> > > Here, we would like to elaborate more about the baseline comparisons.
> > > - **About the comparison settings.** Deliberate alignment firstly designs the OpenAI safety policies, and then prompts a reasoning model to generate CoT containing specific rules as SFT training data. During the procedure, a reward model is used as the judge to filter the training data. In the following RL phase, the model is also used to provide the reward signal. This setting is identical to our `surrogate W/` ablation, in which the guard model is used for both data filtering and the reward model, and the CoT data is generated via distillation. Therefore, it is fair to claim the setting as an acceptable replication of DA. Besides, the experimental settings in the ablation study are identical to those in the main results. Here we would like to provide the detailed results. We will also try to modify the structure to present more results within the 8-page limit.
> > >
> > > **Performance comparison on Safety Benchmarks and Over-refusal Benchmarks across three model sizes. The last column in each section denotes the average score. All results are in percentage (\%).**
> > >
> > > |ModelSetting|SR|WildJ|JBB|WildC|WildG|**Avg**|XST|WildJ|WildG|OKT|OR|**Avg**|
> > > |---|---|---|---|---|---|---|---|---|---|---|---|---|
> > > |**Qwen3-4B**|
> > > |Surragate (W/)|98.40|91.95|99.00|97.73|97.88|**96.99**|7.20|2.38|2.54|5.00|2.58|**3.94**|
> > > |Surragate (W/o)|98.72|46.80|99.00|63.00|79.84|**77.47**|4.80|1.90|2.43|8.00|14.03|**6.23**|
> > > |GRPO|99.36|97.45|100.0|99.24|99.34|**99.08**|7.20|1.43|2.33|7.67|3.49|**4.42**|
> > > |IFPO|100.00|96.25|100.00|99.04|99.07|**98.87**|7.60|2.38|4.34|9.67|5.99|**6.00**|
> > > |**Qwen3-8B**|
> > > |Surragate (W/)|99.36|95.80|100.0|98.76|98.94|**98.57**|8.40|0.00|2.01|6.67|2.88|**3.99**|
> > > |Surragate (W/o)|99.68|67.25|99.00|82.78|88.99|**87.54**|1.60|3.81|3.28|6.00|16.83|**6.30**|
> > > |GRPO|100.0|94.60|100.0|99.40|98.81|**98.56**|6.40|1.90|2.33|8.33|6.44|**5.08**|
> > > |IFPO|100.00|96.90|100.00|99.05|98.94|**98.98**|7.60|2.38|4.34|9.67|5.99|**4.33**|
> > > |**Qwen3-14B**|
> > > |Surragate (W/)|99.68|92.55|100.0|97.17|98.14|**97.51**|9.60|1.90|2.33|5.00|2.05|**4.18**|
> > > |Surragate (W/o)|100.0|76.05|100.0|88.40|92.44|**91.38**|2.80|8.57|4.34|9.67|13.12|**7.70**|
> > > |GRPO|100.0|95.45|100.0|99.48|98.28|**98.64**|6.80|0.95|3.17|8.67|6.75|**5.27**|
> > > |IFPO|100.00|95.45|100.00|99.22|98.54|**98.64**|7.20|1.43|3.60|9.67|7.28|**5.84**|
> > >
> > >
> > > **Model performance comparison on general reasoning benchmarks. All results are in percentage (%).**
> > >
> > > |ModelSetting|MATH|GPQA-D|MMLU-P|GSM8K|ARC-C|Avg.|H-avg.|
> > > |---|---|---|---|---|---|---|---|
> > > |**Qwen3-4B**|
> > > |Surragate (W/)|89.54|45.96|60.91|93.43|89.16|75.80|**88.46**|
> > > |Surragate (W/o)|96.44|48.99|62.55|97.44|93.86|79.86|**83.11**|
> > > |GRPO|85.64|49.49|62.15|95.92|89.25|76.49|**89.21**|
> > > |IFPO|91.20|52.53|63.64|97.40|90.53|79.06|**89.82**|
> > > |**Qwen3-8B**|
> > > |Surragate (W/)|84.88|40.91|64.49|94.79|92.75|75.56|**88.77**|
> > > |Surragate (W/o)|97.82|56.06|66.90|99.14|94.20|82.82|**87.80**|
> > > |GRPO|92.60|51.52|68.34|97.23|89.68|79.87|**90.36**|
> > > |IFPO|94.31|53.03|68.99|97.62|90.02|80.79|**91.09**|
> > > |**Qwen3-14B**|
> > > |Surragate (W/)|90.89|53.54|70.14|97.60|94.28|81.29|**90.93**|
> > > |Surragate (W/o)|98.41|63.64|71.69|98.58|95.31|85.53|**89.63**|
> > > |GRPO|91.88|53.54|71.54|98.35|93.60|81.78|**91.12**|
> > > |IFPO|92.46|58.59|70.79|98.27|94.62|82.95|**91.43**|
> > >
> > > - **Concerns about the IFPO.** We appreciate the idea that adding additional perspectives for comparisons may make IFPO more convincing. Actually, the method is not designed for improving efficiency, and according to the results in our rebuttal to `reviewer vYJe`, our method achieves a comparative time duration with baseline methods. Therefore, we could only claim that our method hardly introduces an overall time overhead.
> > >
> > > To summarize, we sincerely thank you for your detailed and valuable feedback. **The core message we aim to convey through our manuscript lies in our proposed framework: rather than directly learning safety rules, we find it better to prefill the rules via context management, as modern models have already been optimized for contextual reasoning.** We fully respect your perspective, while we sincerely hope our explanations in this rebuttal could highlight the intuition and be helpful for your evaluation.

---

### Decision · Program_Chairs · 2026-04-30

**Decision:**

Accept (regular)

**Comment:**

This paper proposes a training and inference pipeline for improving LLM safety while mitigating over-refusal behavior and preserving general reasoning performance. After the rebuttal phase, the main concerns focused on the exact scope of the “utility” claim, the limited novelty of the work (Reviewer eutU and Reviewer HCnu) and empirical justification of the proposed method (Reviewer iWm3 and Reviewer HCnu). However, the authors engaged actively in rebuttal and addressed these concerns by clarifying the definition of utility, adding supplementary reasoning and cross-model results, and providing more ablations and baselines. Three of four reviewers ended at weak accept or borderline accept. Overall, I lean toward accept.